

# Assessing the Feasibility of Using a Neural Network to Filter OCO-2 Retrievals at Northern High Latitudes

Joseph Mendonca[1], Ray Nassar[1], Christopher W. O'Dell[2], Rigel Kivi[3], Isamu Morino[4], Justus Notholt[5], Christof Petri[5], Kimberly Strong[6], and Debra Wunch[6]

[1]Environment and Climate Change Canada, Toronto, ON, Canada
[2]Cooperative Institute for Research in the Atmosphere, Colorado State University, Fort Collins, CO, USA
[3]Finnish Meteorological Institute, Sodankylä, Finland
[4]National Institute for Environmental Studies (NIES), Tsukuba, Japan
[5]Institute of Environmental Physics, University of Bremen, Bremen, Germany
[6]Department of Physics, University of Toronto, Toronto, ON, Canada

*Correspondence to*: Joseph Mendonca (joe.mendonca@canada.ca)

**Abstract.** Satellite retrievals of $XCO_2$ at northern high latitudes currently have sparser coverage and lower data quality than most other regions of the world. We use a neural network (NN) to filter OCO-2 B10 bias-corrected $XCO_2$ retrievals and compare the quality of the filtered data to the quality of the data filtered with the standard B10 quality control filter. To assess

the performance of the NN filter, we use Total Carbon Column Observing Network (TCCON) data at selected northern high latitude sites as a truth proxy. We found that the NN filter decreases the overall bias by 0.25 ppm (~50%), improves the precision by 0.18 ppm (~12%), and increases the throughput by 16% at these sites when compared to the standard B10 quality control filter. Most of the increased throughput was due to an increase in throughput during the spring, fall, and winter seasons. There was a decrease in throughput during the summer, but as a result the bias and precision were improved during the summer

months. The main drawback of using the NN filter is that it lets through fewer retrievals at the highest latitude Arctic TCCON sites compared to the B10 quality control filter, but the lower throughput improves the bias and precision.

## 1 Introduction

Northern high latitude regions are undergoing considerable changes related to climate change. The Arctic has seen the annual average temperature increase three times more than the global annual average (Stocker et al., 2013). The Boreal forest (an

important driver of the $CO_2$ seasonal cycle) has seen its growing season lengthen due to climate change (Pulliainen et al., 2017), with an increase in the frequency and severity of forest fires (Seidl et al., 2017). Permafrost soils of the northern high latitudes are a large carbon reservoir and some fraction of this carbon is vulnerable to release as $CO_2$ and $CH_4$ as the climate warms (Schuur et al., 2015). Changes in the carbon cycle will impact the climate, which in turn will impact the carbon cycle. Understanding how the carbon cycle is changing at Boreal and Arctic latitudes, including this feedback loop, will be key to

predicting future climate change.



In situ atmospheric measurements of $CO_2$ can be used to study how the carbon cycle is changing. However, cost and logistical challenges present barriers to establishing measurement sites at high northern latitudes, limiting the amount of information available about the carbon cycle in the Arctic and Boreal regions. Remote sensing measurements from space can be used to

complement coverage to the current in situ networks (Olsen and Randerson, 2004). Current satellite missions such as the Greenhouse Gases Observing Satellite (GOSAT) (Yokota et al., 2009) and the Orbiting Carbon Observatory 2 (OCO-2) (Crisp et al., 2004) record solar absorption spectra reflected off the Earth's surface which are used to retrieve column-averaged dry-air mole fractions of $CO_2$ ($XCO_2$), giving regional information on atmospheric $CO_2$. These data can be used to learn about the carbon cycle but require low bias and high precision to be useful (Rayner and O'Brien, 2001).


The density of satellite retrievals of $XCO_2$ from current missions is limited by the amount of available sunlight and the inability to measure through clouds. At high latitudes there is less sunlight available during the colder seasons decreasing the number of spectra obtained when compared to the mid-latitudes. Furthermore, filtering and bias correction schemes are optimized for mid-latitudes where more validation datasets are available. This has led to a filter that removes a larger fraction of the high-

latitude data than data at mid-latitudes. Scenes with snow are also filtered out because they are thought to be problematic for the retrievals, which decreases the throughput during the colder seasons. In order to improve the quality and throughput of retrievals at high latitudes, in this study we focus on using high-latitude validation $XCO_2$ retrievals to improve the filtering of Northern high-latitude OCO-2 bias corrected $XCO_2$ retrievals.

The study by Jacobs et al. (2020) showed that making modifications to the quality control filtering scheme and bias correction used by OCO-2, one can increase the throughput of OCO-2 retrievals (data version B9) (Kiel et al., 2019; O'Dell et al., 2018) in the Boreal region. This was done by changing limits on the features used in the quality control scheme created in O'Dell et al. (2018). These changes were validated by comparing OCO-2 $XCO_2$ retrievals (Kiel et al., 2019; O'Dell et al., 2018) coincident to $XCO_2$ retrievals from ground-based solar absorption spectra made by remote sensing instruments used by the

Total Carbon Column Observing Network (TCCON) (Wunch et al., 2011a).

Machine learning algorithms are useful for pattern recognition in complex data sets. Mandrake et al. (2013) was the first study to demonstrate the use of machine learning (using a genetic algorithm) to filter ACOS-GOSAT retrievals and multiple versions of the OCO-2 retrievals using warn levels. There is potential to apply different machine learning algorithms to the Northern

high-latitude OCO-2 data set in order improve the bias, precision, and throughput.

In this study, we investigate the feasibility of using a simple neural network to filter the current OCO-2 data version (B10) (Osterman et al., 2020) $XCO_2$ retrievals at Northern high latitudes. Section 2 outlines the coincidence criteria between OCO-2 and TCCON retrievals and an explanation of how the retrieved $XCO_2$ is adjusted for different averaging kernels and a priori

information when comparing OCO-2 to TCCON. Section 3 describes the architecture of the neural network and how it is



trained to filter the OCO-2 bias-corrected $XCO_2$ retrievals. In Section 4, the NN filtered OCO-2 retrievals are compared to the B10 quality control (qc_flag) retrievals to assess the performance of the NN filter. Finally, we discuss results of the study and future work to improve the NN filtering.

## 2 Coincidence Criteria

The OCO-2 satellite was launched on July 2, 2014 and has been making measurements since mid-September 2014. The instrument on board the satellite is a three channel, imaging, grating spectrometer that records spectra of reflected sunlight in three spectral bands centered at 0.765 µm, 1.62 µm, and 2.04 µm. These spectra are processed using a "full-physics" retrieval algorithm that retrieves a profile of $CO_2$ (which is used to calculate $XCO_2$) and other geo-physical information. In this study we use OCO-2 data that has been processed using the B10 version of the full-physics retrieval algorithm, with the retrieval

output and sounding information contained in the B10 lite files (Osterman et al., 2020).  All soundings used in the study were recorded from September 2014 to July 2020.

TCCON is a global network of ground-based Fourier Transform Infrared (FTIR) spectrometers that record direct solar absorption spectra. The high-resolution spectra are processed using the GGG2014 retrieval algorithm which scales the a priori

profile of the gas of interest until the spectrum calculated by forward model best matches the spectrum recorded by the FTIR (Wunch,  et al., 2015). GGG2014 retrieves $XCO_2$, $XCH_4$, $XCO$, $XN_2O$, $XHF$, and $XH_2O$ from a single spectrum. Selected $XCO_2$ TCCON retrievals made in the Boreal and Arctic regions were used as a truth proxy to compare to OCO-2 retrievals. The TCCON sites used in this study and the date range of the data are East Trout Lake, Canada (et) (Wunch et al., 2018) from Oct 2016 to June 2020, Eureka, Canada (eu) (Strong et al., 2019) from September 2014 to July 2020, Park Falls, USA (pa)

(Wennberg et al., 2017) from September 2014 to June 2020, Bremen, Germany (br) (Notholt et al., 2019a) from September 2014 to August 2018, Białystok, Poland (bi) (Deutscher et al., 2019) from September 2014 to August 2018, Sodankylä, Finland (so) (Kivi et al., 2014 and Kivi and Heikkinen, 2016) from September 2014 to November 2019, Ny Ålesund, Spitzbergen Norway (sp) (Notholt et al., 2019b) from September 2014 to August 2018, and Rikubestu, Japan (ri) (Morino et al., 2018) from September 2014 to September 2019. Fig. 1 shows the location of all the TCCON sites used in this study. All TCCON

spectra were processed using the GGG2014 algorithm (Wunch, et al., 2015) to retrieve $XCO_2$ and other gases of interest. Data were filtered for standard FLAG = 0, and additionally $XHF <= 150$ ppt, and $XCO <= 125$ ppb.

Filtering for $XHF <= 150$ ppt was done to avoid the impact of the polar vortex on the TCCON retrievals. Arctic sites such as Eureka and Ny Ålesund routinely record solar absorption spectra while under polar vortex conditions during the spring months.

In some years the polar vortex can reach as far south as $40^{\circ}$ N (Whaley et al., 2013). Boreal sites such as East Trout Lake have recorded solar spectra under polar vortex conditions but on fewer days than at the Arctic sites. Since the GGG2014 retrieval algorithm does a profile scaling retrieval (Wunch et al., 2015) it relies on good knowledge of the shape of the profile of the





gases of interest. The GGG2014 profiles are built without knowledge of the impact of the polar vortex on the shape of the profiles. When $XCO_2$ is retrieved from spectra measured through polar vortex conditions, the shape of the a priori profile

generated by the GGG2014 retrieval algorithm will likely be incorrect. This is less of an issue for OCO-2 retrievals because OCO-2 performs a profile retrieval (O'Dell et al., 2018).

The TCCON sites used in this study have no direct influence due to anthropogenic pollution but are still influenced by biomass burning plumes. At sites like East Trout Lake, major enhancements in XCO over background levels are measured, typically in

late summer when measurements are made through forest fire plumes. Even a remote Arctic site like Eureka sees forest fire plumes during the summer months (Viatte et al., 2013). In an attempt to avoid a situation where a coincident TCCON measurement is influenced by a plume and the OCO-2 measurement is not, we filter any TCCON measurement where XCO is elevated above the value of ~150 ppb or more.

We use the B10 "lite" OCO-2 data product (Osterman et al., 2020), where the $XCO_2$ values have been corrected for various biases, such as footprint-to-footprint biases and biases that are dependent on features of the atmosphere, surface, or retrieval algorithm. OCO-2 $XCO_2$ data are also scaled by a global offset term that was derived using the OCO-2 target mode retrievals coincident with TCCON retrievals (Osterman et al., 2020). In our study, we use all OCO-2 spectra that are coincident with the TCCON spectra acquired in nadir and glint modes over land. The coincidence criteria are: the distance of an OCO-2

measurement must be <= 150 km of a TCCON station, the temperature difference between the TCCON and OCO-2 temperature profiles at 700 hPa must be <= absolute value of 2K (Wunch et al., 2011), and the time difference between the TCCON and OCO-2 measurements must be <= 2hrs to avoid the impact of the $XCO_2$ diurnal cycle.

To compare TCCON and OCO-2 retrievals, one has to take into account that GGG2014 and the OCO-2 retrievals obtain

information about atmospheric $CO_2$ from different spectral regions (which have peak sensitivity at different altitudes), and use different a priori information. To adjust the OCO-2 bias-corrected $XCO_2$ retrievals to take into account the a priori profile used in the GGG2014 retrieval, the following formula is used:

$$XCO_{2adj}^{OCO-2} = XCO_{2bc}^{OCO-2} + \sum_j h_j^{OCO-2}(1 - a_j^{OCO-2})(\vec{x}^{TCCON} - \vec{x}^{OCO-2})_j, \qquad (1)$$

where $XCO_{2bc}^{OCO-2}$ is the original bias-corrected $XCO_2$ value found in the lite files, $h_j^{OCO-2}$ is the OCO-2 pressure weighting

vector, $a_j^{OCO-2}$ is the OCO-2 total column averaging kernel vector, $\vec{x}^{OCO-2}$ is the $XCO_2$ a priori profile used in the OCO-2 retrieval and $\vec{x}^{TCCON}$ is the $XCO_2$ a priori profile used in the GGG2014 retrieval but interpolated onto the OCO-2 retrieval pressure grid.

OCO-2 retrieves information about $CO_2$ from the strong $CO_2$ band (centered at 2.04 µm) and the weak $CO_2$ band (centered at

1.62 µm) (O'Dell et al., 2018). TCCON retrieves information from two weak $CO_2$ bands, centered at 1.62 and 1.57 µm, (Wunch





et al., 2011b) but not in the strong $CO_2$ band. This results in the OCO-2 retrievals, having different vertical sensitivities compared to the TCCON retrievals. To take this into account when comparing OCO-2 and TCCON retrievals the following formula is used to adjust the TCCON retrieved $XCO_2$:

$$XCO_{2adj}^{TCCON} = XCO_{2apriori}^{TCCON} + \sum_j h_j^{OCO-2} a_j^{OCO-2} (\gamma \vec{x}^{TCCON} - \vec{x}^{TCCON})_j, \qquad (2)$$

where $XCO_{2apriori}^{TCCON}$ is the integrated a priori profile used in the GGG2014 retrieval, $h_j^{OCO-2}$ is the OCO-2 pressure weighting vector, $a_j^{OCO-2}$ is the OCO-2 total column averaging kernel vector, $\vec{x}^{TCCON}$ is the $XCO_2$ a priori profile used in the GGG2014 retrieval, and $\gamma$ is the TCCON $XCO_2$ value divided by $XCO_{2apriori}^{TCCON}$. Ideally $\gamma$ should be the scaling factor determined by the GGG2014 retrieval, but this value does not take into account the airmass dependence correction and aircraft calibration factor applied in post processing to the retrieved $XCO_2$. The vectors $a_j^{OCO-2}$ and $\vec{x}^{TCCON}$ have been interpolated onto a 20-layer

pressure grid using the surface pressure measured at the TCCON site.

The bias between coincident TCCON and OCO-2 retrievals is calculated by taking the difference between $XCO_{2adj}^{OCO-2}$ (Eq. 1) and $XCO_{2adj}^{TCCON}$ (Eq. 2) and resulting in:

$$XCO_2^{Diff} = XCO_{2adj}^{OCO-2} - XCO_{2adj}^{TCCON}. \qquad (3)$$

**3 Neural Network Architecture and Training**

To filter the OCO-2 data, we use a three-layer neural network (NN) that consists of an input layer, a hidden layer, and an output layer. The design of the NN is based on the book by Nielsen (2015). The input layer is the value of the features of the OCO-2 retrievals that are given in the B10 lite files. Table 1 lists all the features used by the NN. The hidden layer contains the "neurons" where the calculations are done. Each input is connected to a neuron by a weight. The calculation for single

neuron ($N$) in a NN with $k$ neurons is given by:

$$N^k = \sum_{i=1}^n w_i^k I_i + b^k, \qquad (4)$$

where $I_i$ is the value of feature $i$, $w_i^k$ is the weight on feature $i$ for neuron $k$, and $b^k$ is the bias associated with neuron $k$. There is a total of 37 neurons which is the total number of features plus one. An activation function is commonly applied to the neuron in order to introduce some non-linearity into the neuron calculation and make sure that small changes in the values of

$w_i^k$ and $b^k$ result in small changes in the final output values when training the NN (Nielsen, 2015). The sigmoid function:

$$f(N^k) = \frac{1}{1 + e^{-N^k}}, \qquad (5)$$

is used as the activation function. Each neuron is linked to the final output value by a weight ($w_k$). The output value is given by:

$$\hat{Y} = f\left(\sum_{i=1}^k w_k f(N^k) + b\right), \qquad (6)$$



where $b$ is the offset and everything else is as described as before.

Applying the sigmoid activation function in Eq. 6 ensures that $\hat{Y}$ will have a value between 0 and 1. This is useful for binary classification, which in this case we would use the NN to classify the OCO-2 retrieval as either "good" or "bad" by equating a calculated value that is close to 0 as "good" and a calculated value that is close to 1 as "bad".


For the NN to work, the values of $w_i^k$, $b^k$, $w_k$, and $b$ need to be determined. This was done by using a subset of the OCO-2 coincident retrievals to train the NN. The coincident data set consists of co-located OCO-2 soundings at the following TCCON sites: East Trout Lake (et), Eureka (eu), Bremen (br), Białystok (bi), Sodankylä (so), Ny Ålesund (sp), and Rikubestu (rj). We withhold the Park Falls (pa) data set so that it can be a completely independent source of validation. The coincident data were

split into three datasets: training, testing, and validation. For the training and testing data, 20% of the data were randomly selected to go into each data set, with the remaining 60% used for validating the results. In order to train the NN, one needs to know the input values of the training data set (which are the values of the features in the B10 lite files) and the expected output value ($Y$). The expected output value was set to $Y = 0$ if the difference between a coincident OCO-2 and TCCON retrieval is $<= \pm 2.5$ ppm and set to $Y = 1$ if the difference between the retrievals is $> 2.5$ ppm. Fig 2a shows the histogram of the difference

between coincident OCO-2 and TCCON retrievals as well as the boundaries separating data into expected values of 0 and 1. All data between the red dashed lines was set to $Y = 0$ (or "good") and set to $Y = 1$ (or "bad") if outside of the boundary.

To achieve the best results when training the NN, we standardize the values of the input features so that each feature has a similar range of values. This is helpful because the features have different units and orders of magnitude, and if left as is the

NN will place much more importance on features that have large absolute values than other features with smaller values. To standardize the input features the following formula is used:

$$z_i = \frac{I_i - \mu_i}{\sigma_i}, \tag{7}$$

where $I_i$ is as before, $\mu_i$ is the mean of $I_i$ values from the training data set, and $\sigma_i$ is the standard deviation of the $I_i$ values from the training data set. This means that $z_i$ is used in Eq. 4 instead of $I_i$. The supplementary excel file (sheet Standardize values)

contains the $\mu_i$ and $\sigma_i$ for each of the features to be used to standardize the data before inputted into the NN.

To determine the values of $w_i^k$, $b^k$, $w_k$, and $b$, they are initially set randomly to be between a value of $\pm 1$. Using the training data set, $\hat{Y}$ is calculated for all the data using the initial values of $w_i^k$, $b^k$, $w_k$, and $b$. The performance of the NN is then determined by comparing the calculated value ($\hat{Y}$) to the expected output value ($Y$) using the log loss entropy cost function:

$$C = \frac{-1}{n} \sum_i^n Y^i \log(\hat{Y}^i) + (1 - Y^i) \log(1 - \hat{Y}^i), \tag{8}$$





where $n$ is the total number of OCO-2 retrievals in the training data set. If $\hat{Y} = Y$ then $C$ will equal zero, meaning the values of $w_i^k$, $b^k$, $w_k$, and $b$ are set to the best values that perfectly determine whether an OCO-2 retrieval is good or bad. This is unlikely to happen for the initial values of those variables since they are set randomly, so $C$ will be $> 0$. To minimize the value of $C$, the values of $w_i^k$, $b^k$, $w_k$, and $b$ are adjusted. The adjustments are done by taking the partial derivative with respect to the cost

function (i.e., $\frac{\partial C}{\partial w_i^k}$, $\frac{\partial C}{\partial b^k}$, $\frac{\partial C}{\partial w_k}$, and $\frac{\partial C}{\partial b}$). In principle, this should be iterated until $C = 0$ but in practice, the classification of the training data setup is not perfect. The assumption made when setting up the classification of the training data is that if -2.5 ppm $< XCO_2^{Diff}$ (Eq. 3) $< 2.5$ ppm then it is a good OCO-2 retrieval but this might not be true. It could be the case that the OCO-2 retrieval has adjusted parameters as much as possible to achieve the best possible fit to the measured spectra, but that the retrieved parameters deviate from the true values while still providing an integrated profile that is close to the TCCON

$XCO_2$. This retrieval would be mis-classified as good, so the cost function will never reach 0.

To stop training the NN, a few cutoffs were placed: the maximum number of iterations is 5000 or the accuracy between the training and testing data $< 3\%$. When training the NN, the accuracy of the training data and the testing data is calculated on each iteration and compared. Since the data were set up in a binary classification (i.e., 0 or 1), on each iteration, if a calculated

value was $<= 0.1$ (unitless) the classification was set to 0 and $> 0.1$ the classification was set to 1.0. These classification values were compared to the expected classification value on each iteration to get the accuracy of the training and testing data sets. The testing dataset is not used when determining the values of $w_i^k$, $b^k$, $w_k$, and $b$, rather it is used as an independent data source to make sure that the NN is not overfitting the training data. The derived values of $w_i^k$, $b^k$, $w_k$, and $b$ can be found in the supplementary excel file with values of $w_i^k$ in sheet w1, $b^k$ in sheet b1, $w_k$ in sheet w2, and $b$ in sheet b2.


Figure 3 shows the $XCO_2^{Diff}$ as a function of the value calculated by the NN for all three data sets. Fig. 3a shows that the OCO-2 retrievals with calculated values close to 0 have the smallest spread in $XCO_2^{Diff}$, while calculated values close to 1 have the largest spread. This pattern is seen in all three of the datasets. The density plot shown in Fig. 3b confirms that for most of the data the calculated values are $<= 0.1$. There is no clear separation of data (i.e. good retrievals $<= 0.1$ and bad

retrievals $>= 0.9$) as one would expect from a binary classifier. This could be because there are many combinations of feature values that can lead to a bad retrieval.  Another possibility is that most of the training data were classified as good and so there are more examples of good retrievals than bad retrievals to learn from.

## 4 Validation

To validate the NN filtering, the validation data set was separated into two data sets. One data set was the OCO-2 bias-corrected

$XCO_2$ values filtered using the NN filter and the other was filtered using the B10 qc_flag=0. Since the validation data set was not used in the training of the NN, it is an independent data set kept aside to assess the performance of the NN filter. Table 2





shows the bias, scatter, and number of retrievals for the entire validation data set (All) and at each site when applying either the NN or qc_flag filter. The overall $XCO_2$ bias using the NN filter is half of the qc_flag filter, the scatter has been decreased by 0.18 ppm, and the throughput has been increased by 16%. The NN filter reduces the bias at every site except at Eureka and

Rikubetsu. The precision is better at every site when the NN filter is applied to the validation data. The throughput has increased at every site, when the NN filter is used, except for the Arctic sites (Eureka and Ny Ålesund). Park Falls data was not used to train the NN filter because it is slightly outside of the Boreal domain and it is used as a completely independent data set to validate the NN filter. When the NN filter is applied to Park Falls data, the bias remains the same, the precision decreases by 0.09 ppm, and the throughput increases by ~20%.


The reduction in throughput at the Arctic sites is because the distribution of data at the Arctic sites is different compared to all other sites as shown in Fig. 2b. The peaks of the histograms for the Arctic sites are closer to the boundaries used to classify the training data as "good" or "bad", so almost half of the data is set to "bad" when training the NN. Fig. 4 shows the pass rate for the NN filter given the value of the solar zenith angle (Fig. 4a), sensor zenith angle (Fig. 4b) and altitude standard deviation

(Fig. 4c). In all three plots the data are binned, with the blue dots showing the number of OCO-2 soundings that pass the NN filter divided by the total amount of data multiplied by 100 in each bin. The pink bars are the histogram of OCO-2 soundings coincident with Eureka TCCON data. Fig. 4a shows that the coincident OCO-2 soundings are made at solar zenith angles between 58° to 85°, with the blue dots showing that 30% to 0% of the soundings that have these values pass the NN filter. Similarly Fig. 4b shows that the coincident OCO-2 soundings are made at high sensor zenith angles, which are less likely to

pass the NN filter. Most of the coincident OCO-2 soundings at Eureka are made over land that contains significant topographic variability. Fig. 4c shows the altitude standard deviation, which is the standard deviation of the elevation (in meters) of the field of viewing of the sounding. The plot shows that at an altitude standard deviation of ~ 50 m only 30% of the soundings pass the NN filter. The combination of high airmass and variable topography decreases the throughput at Eureka.

For further validation, the seasonal bias, scatter, and number of retrievals that pass the filters at each site is compared. Fig. 5 shows the bias at each site for spring, summer, fall, and winter when the NN filter is applied to the validation data (solid bars) and also when the qc_flag filter (dashed bars) is used on the same validation data set. For most sites and seasons, the magnitude of the biases for the two different filtering schemes are similar, although in most cases the NN filter has a lower absolute bias compared to the qc_flag filter. The NN filter significantly improves the bias at Sodankylä and Rikubetsu during spring, Ny

Ålesund during summer, and Rikubetsu and Bremen during winter. Both the NN filter and the qc_flag show there is a positive bias between OCO-2 and TCCON in summer. The NN filter is able to reduce this summer bias but it still remains. At Park Falls the bias between the two filters is similar for the different scenes, with the qc_flag showing a lower bias in summer and the NN filter decreasing the bias in winter.





Figure 6 shows the precision at each site for spring, summer, fall, and winter when the different filters are applied to the validation data. The precision is very similar (i.e., within 0.2 ppm) for most sites during the different seasons. The NN filter improves the precision (by more than 0.2 ppm) at Rikubetsu during spring, Eureka and Ny Ålesund during summer, and Białystok and Rikubetsu during fall. However, the qc_flag filter has a much better precision at Sodankylä during spring when compared to the NN filter.


Figure 7 shows the number of retrievals that pass each filter for the different sites during spring, summer, fall and winter. At most sites, the NN filter lets through more retrievals compared to the qc_flag filter during spring, fall, and winter. In summer the qc_flag filter has a slightly higher throughput compared to the NN filter at most sites. This decrease in throughput during summer helps improve the bias and precision as seen in Figs. 5b and 6b. There is a significant increase in throughput at East

Trout Lake during spring with the NN filter, and it even produces some retrievals in winter. At Park Falls the throughput has increased in spring, fall, and winter but is significantly decreased during summer. The decrease in summer is because the NN filter is trained on data that show a bias during summer, which it decreases by filtering out more data compared to the qc_flag filter. Even though Park Falls is not in the Boreal domain, its scene type (forest) is similar to East Trout Lake. The NN has no information on time of year, but it does have information on the surface type through the albedo values, which change due to

the time of year. It's most likely that what the NN learned from the East Trout Lake data is influencing how the NN filters the data at Park Falls.

Some of the increase in throughput with the NN filter during spring, fall, and winter can be explained by the fact that the qc_flag filter tries to filter out spectra that have been recorded over snow scenes (Osterman et al., 2020). The snow_flag, found

in the B10 lite files is used to indicate the presence of snow in the scene. We applied this snow_flag to the NN filtered data to see if the NN filter removes all soundings over snow. From the validation data set, 3219 retrievals have snow_flag = 1, with 785 of those retrievals passing the NN filter. This means that the NN filter passes about 24% of the OCO-2 soundings made over snow. This is much lower compared to the general case (all scenes) where greater than 40% of the data pass both the NN and qc_flag filters. The bias over snow scenes compared to TCCON from all the retrievals that pass the NN filter in the

validation data set is $0.13 \pm 1.44$. Since the precision is lower over snow, it makes sense that the throughput over snow is lower compared to the general case. At Park Falls 1032 soundings that pass the NN filter, with 727 in winter and 302 in spring, are snow scenes. So a significant amount of throughput during winter at Park Falls are made over snow scenes. The bias of snow scenes at Park Falls was found to be $0.12 \pm 1.41$.

The NN filter was applied to all OCO-2 B10 data at latitudes greater than $45^\circ$ N to determine the throughput in the Boreal and Arctic regions. Fig. 8 shows the percent difference (NN minus qc_flag, divided by qc_flag, and multiplied by 100) between the number of soundings that pass the filters. The maximum value for the percent difference was capped at 100%. The throughput with the NN filter is greater than the qc_flag filter in spring and winter, while the throughput with the qc_flag filter



is greater than the NN filter in summer and fall. This is consistent with what is seen at the individual TCCON sites. Over
Greenland the throughput has increased with the NN filter regardless of season because qc_flag filter removes all data over
Greenland with the snow_flag filter. During fall, the throughput has increased at greater than 70° N, because the NN filter is
letting through soundings that were recorded over snow scenes.

## 5 Discussion and Conclusions

In this study, a neural network was used to filter the OCO-2 bias-corrected $XCO_2$ data collected near northern high-latitude
TCCON stations as described in Section 3. The performance of the NN filter was assessed by comparing the bias, precision
and throughput to the quality control filtered data. There was an improvement in the bias, precision, and throughput both
overall and at most sites, as well as improvements in the bias in different seasons. However, the NN filter decreases the
throughput at Eureka because it finds that OCO-2 soundings made at high solar zenith angles, high sensor zenith angles, and
over topography are problematic.


This study shows the potential of using a neural network to filter OCO-2 retrievals that could be useful in future filtering
schemes for OCO-2 or other satellite missions. However, there are potential drawbacks to the methodology presented in this
study. In this study, we focus on data near northern high-latitude TCCON stations and so do not sample globally representative
ranges of surface properties or airmasses. Fig. 1 shows the limited coverage that the TCCON sites provide, with no coverage
over Greenland and most of the Eurasian Boreal region. The effectiveness of the NN filter is dependent on how well the NN
is trained. We train the NN using OCO-2 data coincident with TCCON data, so the NN filter is trained only under atmospheric
conditions observed at the northern high-latitude TCCON sites. We have shown that this way of training the NN is effective
when validated against northern high-latitude TCCON data. When the NN filter was applied to Park Falls data, which was not
used in the training of the NN, we found that the bias was similar to the qc_flag filter, with a decrease of 0.09 ppm in precision,
but a 20% increase in throughput. Although the throughput increased in spring, fall, and winter seasons, it decreased a lot
during summer. The decrease in throughput in summer led to improved bias and precision values at all the TCCON sites used
in the training of the NN, but not at Park Falls. This is because the NN has found a pattern that improves the training data set
which is not as applicable to Park Falls data. The qc_flag filter lets through almost twice as much data compared to the NN
filter during summer with a decrease in precision of only 0.09 ppm compared to the NN filter. The NN filter is sub-optimal at
Park Falls (during summer) and if one applied the NN filter to data that is not similar to northern high-latitude data used to
train the NN it will not be as effective.

The effectiveness of the NN filter is dependent on the data set used to train the NN. In this study we assume that the TCCON
data represents the truth and any bias that we see is in the OCO-2 retrievals. The NN is trying to decrease the bias it sees as
much as possible and if there is a bias in the TCCON data, it will attribute this to a bias in the OCO-2 data and treat that data

as bad. One way to be less influenced by TCCON data is to use a small area approximation, where $XCO_2$ is assumed constant within a small region (O'Dell et al., 2018). While the absolute value of the retrieval cannot be evaluated using a small area analysis, variability within the small area can, and this would vastly increase the dataset size used in the NN, and improve the range of surface properties, atmospheric conditions, and airmasses represented by the training dataset. This small area approach

will be investigated in a future study.

The accuracy of $XCO_2$ observations over the northern high latitudes and the loss of data there due to filtering has been a longstanding issue with OCO-2 and GOSAT data versions to date, which has limited the scientific community's ability to apply their data to investigate important northern high-latitude carbon cycle science questions. This paper demonstrates that a

neural network approach can be used to increase the number of soundings at northern high latitudes, while also improving the bias, precision and throughput depending on the site. Continual efforts at improving northern high-latitude retrievals and filtering will be beneficial not only to current missions, but also to future $XCO_2$ missions like MicroCarb (Pasternak et al., 2017), GOSAT-GW (Kasahara et al., 2020), and $CO_2M$ (Sierk et al., 2019), which will make global observations that include northern high latitudes and even more so for missions under consideration like AIM-North (Nassar et al., 2019), which is

dedicated to observing the Arctic and Boreal atmosphere.

**Data availability**

The OCO-2 B10 lite files were obtained from the NASA Goddard Earth Science Data and Information Services Center (GES DISC; https://oco2.gesdisc.eosdis.nasa.gov/data/OCO2_DATA/OCO2_L2_Lite_FP.10r/, last access: 1 October 2020). TCCON data are available from the TCCON data archive, hosted by CaltechDATA (https://tccondata.org/, last access: 1

August 2020).

**Author Contributions**

JM designed the study, developed the neural network code, analysed the results and wrote the paper. RN provided input into the analysis. DW provided insight into the use of TCCON data, comparisons between TCCON and OCO-2 as well as the overall analysis. CO provided critical analysis of choice of coincident criteria, filtering of OCO-2 data, and insight into OCO-

2 bias correction. RK, IM, JN, CP, KS, and DW are involved with the operation of the TCCON sites, data processing, and use of TCCON data in this study. All authors read and provided feedback to JM on the paper.

**Competing Interests**

Kimberly Strong and Justus Notholt are associate editors of AMT. There are no other conflicts of intrest.





## Acknowledgments

The OCO-2 data were produced by the OCO-2 project at the Jet Propulsion Laboratory, California Institute of Technology, and obtained from the OCO-2 data archive maintained at the NASA Goddard Earth Science Data and Information Services Center. TCCON data were obtained from the TCCON Data Archive, hosted by CaltechDATA, California Institute of Technology. The Rikubetsu TCCON site is supported in part by the GOSAT series project. East Trout Lake support is provided by CFI/ORF and NSERC. The Eureka TCCON measurements were made at the Polar Environment Atmospheric Research

Laboratory (PEARL) by the Canadian Network for the Detection of Atmospheric Change (CANDAC), primarily supported by the Natural Sciences and Engineering Research Council of Canada, Environment and Climate Change Canada, and the Canadian Space Agency. The University of Bremen acknowledges support by the ESA project IDEAS and by DLR within the Sentinal S5P validation projects. We would like to thank Paul O. Wennberg and his team for processing TCCON Spectra at Park Falls and providing that data to the TCCON Data Archive.

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



**Table 1 List of all the features available to the input layer of the neural network and a brief description of the features based on the descriptions found in Osterman et al. 2020.**

| Feature Name | Description of Feature |
| --- | --- |
| Retrieval co2_grad_del | Measure of how much the retrieved profile shape is different compared to the aprior profile. |
| Retrieval dpfrac | Correction of $XCO_2$ due to satellite pointing error. |
| Retrieval eof3_3_rel | Scale factor for of the 3rd eof in the $CO_2$ strong band |
| Retrieval deltaT | Retrieved offset of a priori temperature profile |
| Retrieval h2o_scale | Scale factor of retrieved $H_2O$ column |
| Retrieval aod_oc | Retrieved aerosol optical depth of organic carbon |
| Retrieval aod_water | Retrieved aerosol optical depth of water |
| Retrieval aod_dust | Retrieved aerosol optical depth of dust |
| Retrieval aod_bc | Retrieved aerosol optical depth of black carbon |
| Retrieval aod_strataer | Retrieved aerosol optical depth of stratospheric aerosol |
| Retrieval aod_seasalt | Retrieved aerosol optical depth of sea salt |
| Retrieval aod_sulfate | Retrieved aerosol optical depth of sulfate |
| Retrieval aod_ice | Retrieved aerosol optical depth of ice |
| Retrieval water_height | Retrieved central pressure of cloud water layer relative to surface pressure |
| Retrieval ice_height | Retrieved central pressure of cloud ice layer relative to surface pressure |
| Retrieval dust_height | Retrieved central pressure of dust aerosol layer relative to surface pressure |
| Retrieval albedo_wco2 | Retrieved albedo of weak $CO_2$ band |
| Retrieval albedo_slope_wco2 | Retrieved albedo slope of weak $CO_2$ band |
| Retrieval albedo_quad_wco2 | Retrieved albedo quadratic coefficient of weak $CO_2$ band |
| Retrieval albedo_sco2 | Retrieved albedo of strong $CO_2$ band |
| Retrieval albedo_slope_sco2 | Retrieved albedo slope of strong $CO_2$ band |
| Retrieval albedo_quad_sco2 | Retrieved albedo quadratic coefficient of strong $CO_2$ band |
| Retrieval albedo_o2a | Retrieved albedo of $O_2$ A-band |
| Retrieval albedo_slope_o2a | Retrieved albedo slope of $O_2$ A-band |





| Retrieval albedo_quad_o2a | Retrieved albedo quadratic coefficient of $O_2$ A-band |
|---|---|
| Retrieval rms_rel_o2a | Root mean square residual of $O_2$ A-band relative to continuum signal. |
| Retrieval rms_rel_wco2 | Root mean square residual of weak $CO_2$ band relative to continuum signal. |
| Retrieval rms_rel_sco2 | Root mean square residual of strong $CO_2$ band relative to continuum signal. |
| Sounding altitude_stddev | How much the surface elevation changes within the soundings field of view. |
| Preprocessors max_declocking_sco2 | Estimate of the clocking error in the strong $CO_2$ band. |
| Preprocessors max_declocking_o2a | Estimate of the clocking error in the $O_2$ A-band. |
| Preprocessors max_declocking_wco2 | Estimate of the clocking error in the weak $CO_2$ band. |
| Preprocessors co2_ratio | Ratio of retrieved $CO_2$ column form the strong and weak $CO_2$ bands. |
| Preprocessors h2o_ratio | Ratio of retrieved $H_2O$ column form the strong and weak $CO_2$ bands. |
| solar_zenith_angle | Solar zenith angle of sounding |
| sensor_zenith_angle | Sensor zenith angle of sounding |







**505**   **Table 2 The XCO$_2$ bias and scatter (ppm), and number of OCO-2 retrievals at each TCCON site and overall for the OCO-2 bias-corrected XCO$_2$ after applying either the NN or qc_flag filter to the validation data set. All includes data from every site except for Park Falls (pa).**

| | Neural Network | | B10 qc_flag | |
|---|---|---|---|---|
| Site | Bias ± precision | Number of retrievals | Bias ± precision | Number of retrievals |
| All | 0.25 ± 1.27 | 23429 | 0.52 ± 1.45 | 20198 |
| Eureka (eu) | -0.53 ± 2.35 | 59 | 0.34 ± 2.94 | 634 |
| Ny Ålesund (sp) | 0.87 ± 2.30 | 91 | 2.09 ± 2.65 | 92 |
| Sodanklyä (so) | 0.34 ± 1.23 | 5118 | 0.64 ± 1.30 | 4736 |
| East Trout Lake (et) | 0.01 ± 1.34 | 5261 | 0.44 ± 1.48 | 3186 |
| Białystok (bi) | 0.27 ± 1.16 | 6237 | 0.40 ± 1.18 | 5609 |
| Bremen (br) | 0.42 ± 1.19 | 4066 | 0.85 ± 1.28 | 3672 |
| Rikubetsu (rj) | 0.23 ± 1.39 | 2597 | 0.13 ± 1.70 | 2269 |
| Park Falls (pa) | -0.12 ± 1.27 | 14859 | -0.12 ± 1.18 | 12406 |





Figure 1: Map of the location of all TCCON sites used in this study.



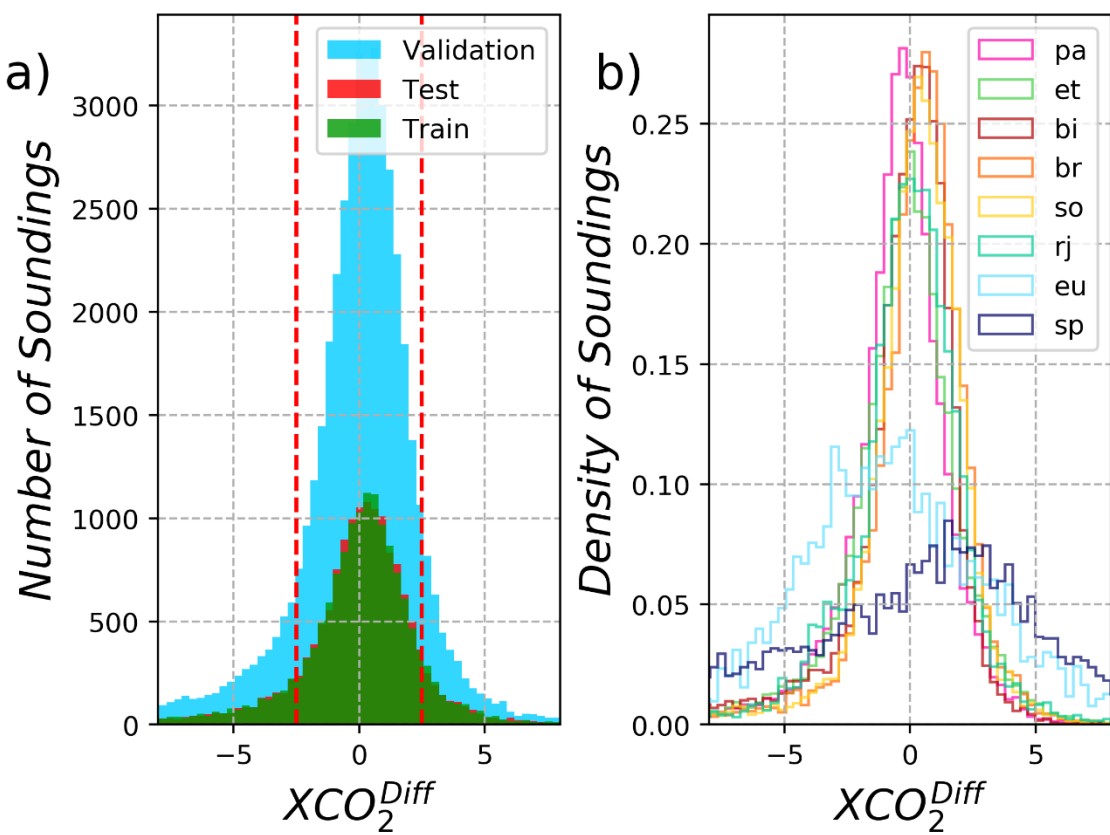

**Figure 2 a) Histogram of the bias between coincident TCCON and OCO-2 retrievals $XCO_2^{Diff}$ (Eq. 3), for the three datasets. The red dashed lines represent the boundary between setting the classification of the data. b) Same as plot 'a)' but shows the density of soundings for each of the sites.**





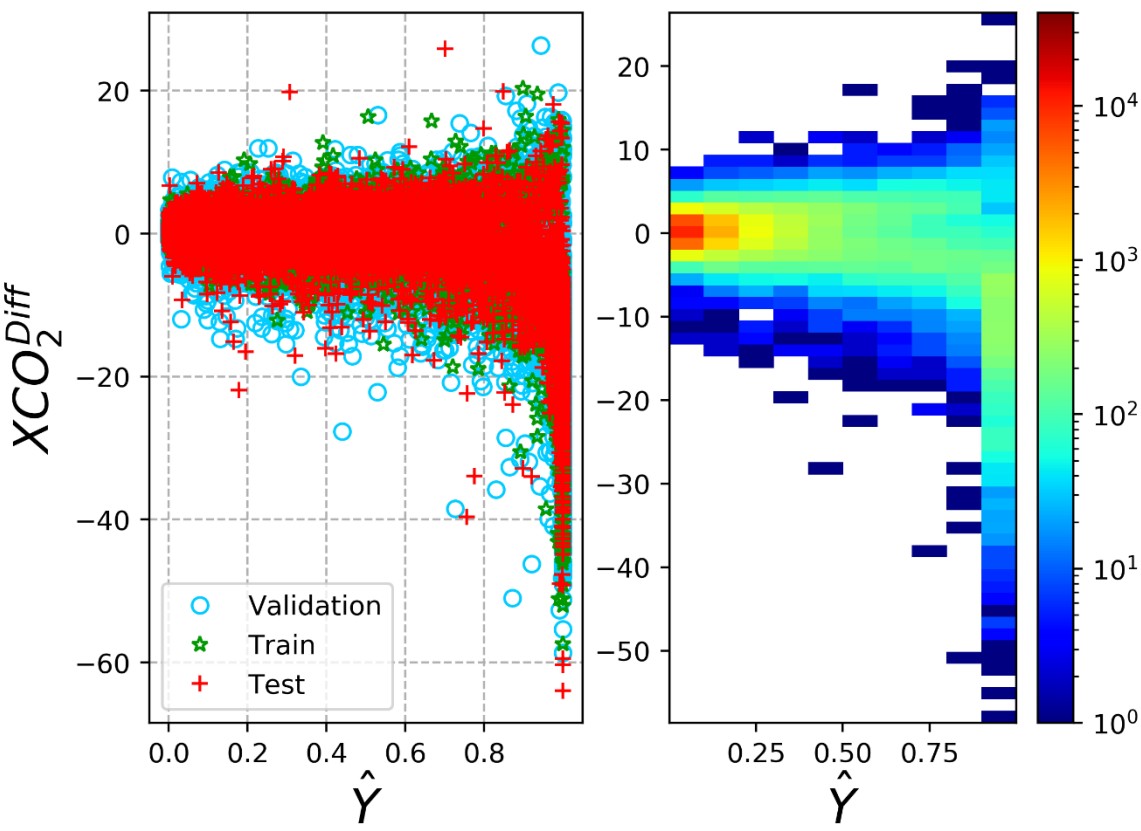


**Figure 3 a) The difference between the coincident TCCON and OCO-2 retrievals ($XCO_2{}^{Diff}$) as a function of $\hat{Y}$ calculated by the NN (after training) for the training, testing, and validation data sets. b) Same as plot a) but shows the density of the all three data sets combined, with the color bar given on a log scale.**



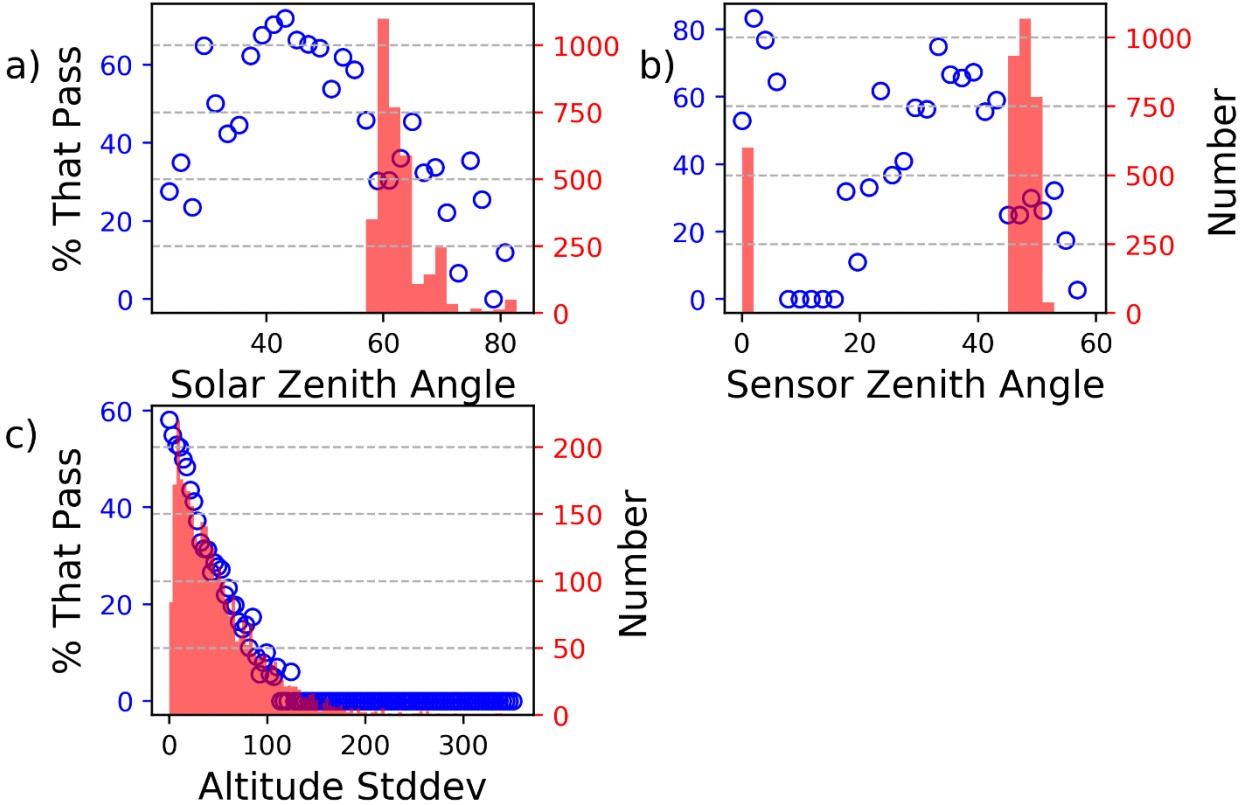


**Figure 4 The percentage of data that passes the NN filter (blue dots) for a given solar zenith angle (a), sensor zenith angle (b) and altitude standard deviation (stddev) (c). The bars are the histograms of the OCO-2 soundings coincident with the Eureka TCCON data.**





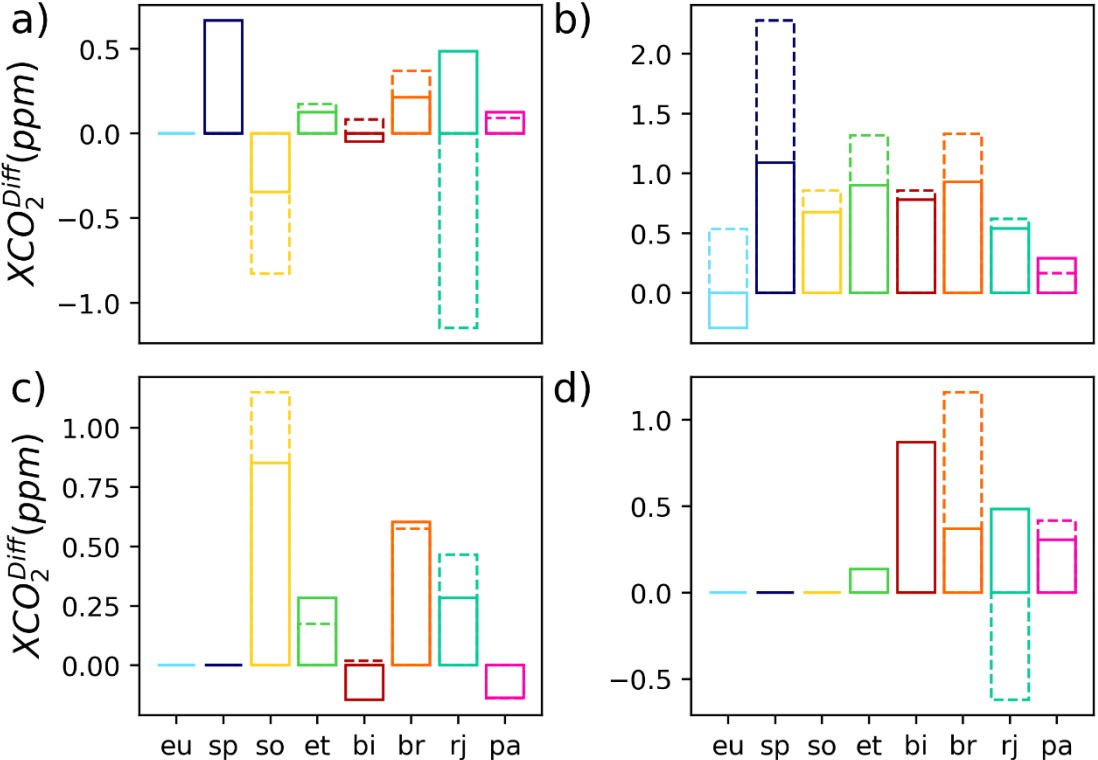

Figure 5 The bias at each site for the different seasons when the NN filter (bars with solid lines) and qc_flag (bars with dashed lines) is used to filter the OCO-2 retrievals in the validation data set. a) Spring (March, April, May). b) Summer (June, July, August). c) Fall (September, October, November). d) Winter (December, January, February). Note the different y-axis ranges for each plot. Note that bars that show a bias of zero are due to no data rather than a bias of zero.



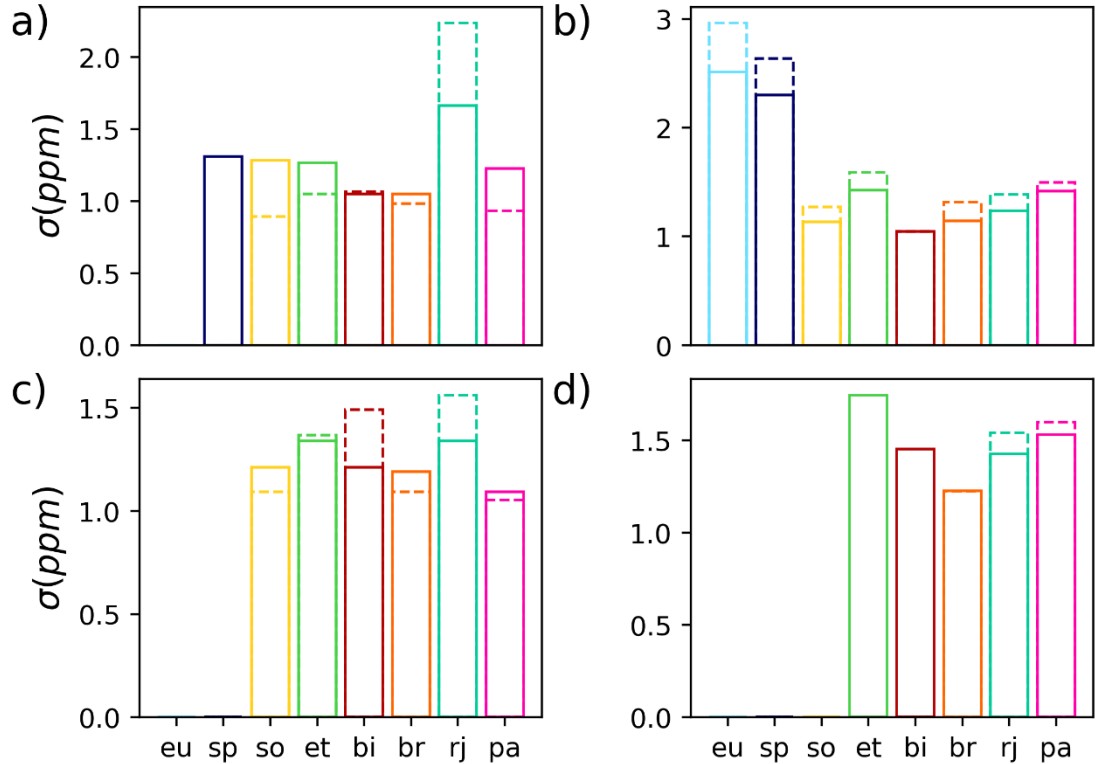

**Figure 6 Same as Figure 5 but shows the precision at each site for each season. a) Spring (March, April, May). b) Summer (June, July, August). c) Fall (September, October, November). d) Winter (December, January, February). Solid bars indicate the NN filter, and dashed bars indicate the original B10 qc_filter.**





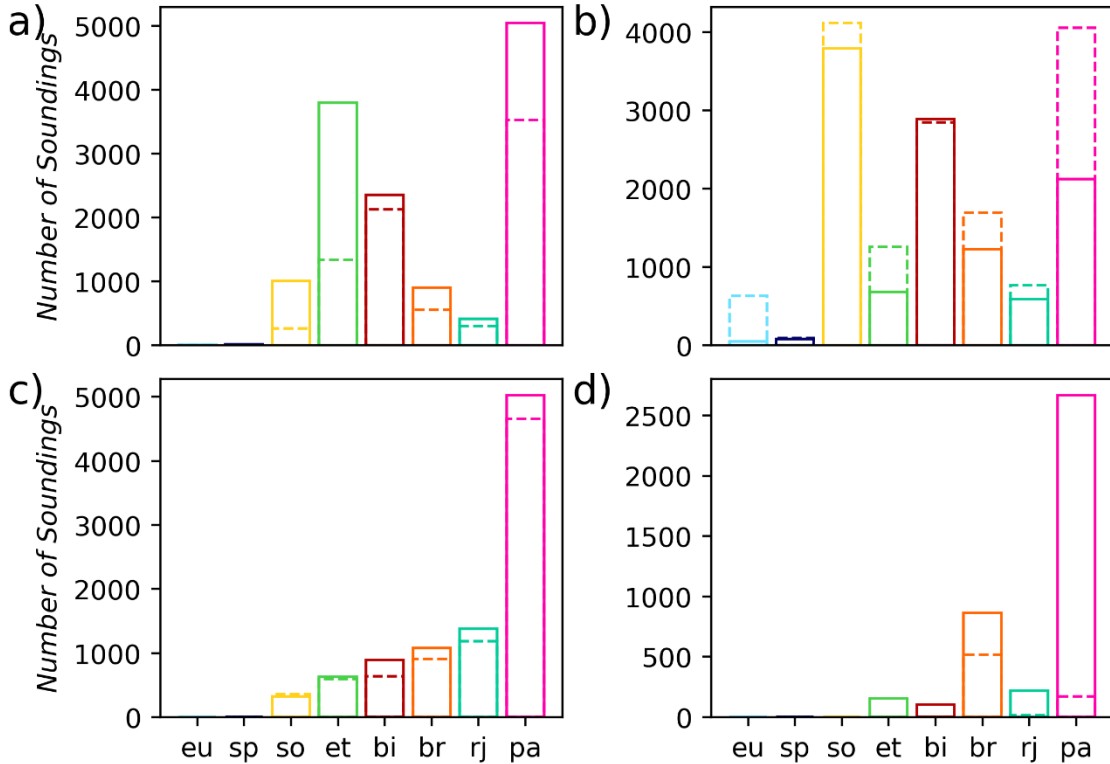

**Figure 7** Same as Figure 5 but shows the number of soundings that pass each filter at each site for the different seasons. **a) Spring (March, April, May). b) Summer (June, July, August). c) Fall (September, October, and November). d) Winter (December, January, February). Solid bars indicate the NN filter, and dashed bars indicate the original B10 qc_filter.**



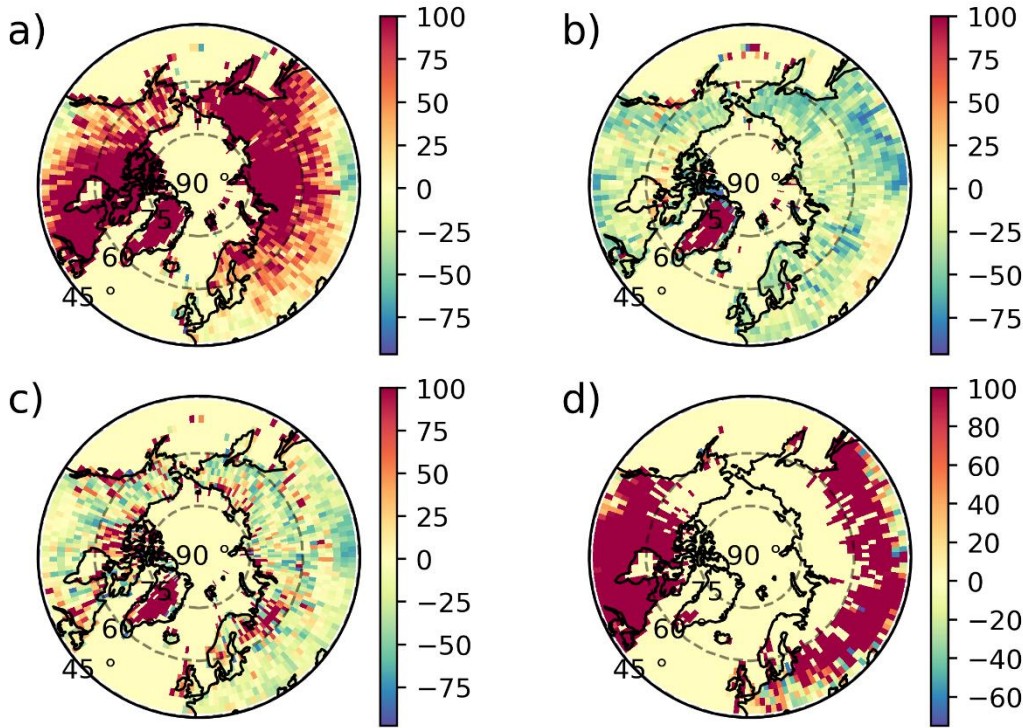

**Figure 8 Polar plots of the percent difference in the number of soundings that pass the NN filter compared to the qc_flag filter for**
**spring (a), summer (b), fall (c), and winter (d). The data have been binned by 2° longitude by 2° latitude.**