# Peer review of "Assessing the Feasibility of Using a Neural Network to Filter OCO-2 Retrievals at Northern High Latitudes"

_Atmospheric Measurement Techniques, 2021_

## Author Comment (AC1)

**Assessing the Feasibility of Using a Neural Network to Filter OCO-2 Retrievals at Northern High Latitudes**

Referee comments are given in Blue.

Responses to comments are given in Black.

In the manuscript highlighted text is the added text and  text is deleted text.

**Response to Referee 1**

We thank the reviewer for the comments on our manuscript. Please see below for our responses.

*Major Comment 1*
It seems a bit of a mismatch to me that NN's would be chosen for an application that is reduced to such a binary distinction; did the authors try or consider other ML techniques such as self-organizing maps or regression trees? Alternatively, the problem could be posed such that the output values are on a continuum, such that values indicate some measure of confidence, e.g., Y=0 if OCO-2=TCCON, Y=0.5 if OCO-2 within +/- 2.5 ppm of TCCON, Y=1 if OCO-2 exceeds +/- 5 ppm, or similar. As the methods are described, it seems as if the training "target" values are strictly 1's and 0's, making the eventual NN outputs in the middle somewhat ambiguous, as referenced below.

I suggest the authors explain their reasoning for the choice of NNs with the binary result and/or discuss alternate setups that might be attempted for future directions.

We did not try self-organizing maps (SOM's) but we did try both k-nearest neighbors (k-nn) and k-means which are also clustering algorithms. The main problem with these clustering algorithms was that the amount of data needed from the coincident data set to train them was so large that the remaining retrievals used for validation could not provide statistically meaningful results when comparing to the OCO-2 qc_flag retrievals. It is possible that SOM's maybe less sensitive to noise in the data (compared to k-nn and k-means) but would likely need a significant portion of the coincident data to train like the other clustering algorithms mentioned.

Decision trees are useful for binary classification of data that is not easily clustered. The draw back to using decision trees is that they overfit the training data. This can be a problem when classifying the OCO-2 retrievals because using $XCO_2{}^{Diff}$ as the proxy to classify the retrieval is not perfect. As briefly mentioned on lines 202-206, it is important to keep in mind that the OCO-2 retrieval algorithm has an imperfect forward model and so geophysical parameters in the retrieval state vector are often tuned (with some constraint) by the retrieval in order to achieve the best possible fit to the measured spectrum. This makes it possible for the retrieval to adjust parameters in order to make up for deficiencies in the forward model, resulting in values that are not representative of the true state of the atmosphere and leading to biases caused by the retrieval. Since OCO-2 is a profile retrieval it can adjust the amount of $CO_2$ at different pressure levels to achieve the best possible fit to the measured spectrum. This may result in a profile with a shape that doesn't represent the true $CO_2$ profile of the atmosphere, but the integrated profile does if for example, some extra $CO_2$ at the surface is balanced out with less $CO_2$ in upper parts of the atmosphere. This makes it possible to achieve a $XCO_2{}^{Diff}$ within ±2.5 ppm but have a bad

retrieval because the retrieved parameters are not representative of the true state of the atmosphere. Therefore, there could be retrievals that meet the $XCO_2{}^{Diff}$ criteria which are classified as "good" for the training data set but could be "bad" retrievals. Please refer to minor comment 3 for more detail on this. Given that $XCO_2{}^{Diff}$ is not a perfect metric to set the classification for the training dataset, overfitting could lead to suboptimal performance when filtering. The ambiguous classification of some retrievals with the NN can be interpreted as a confidence of a given classification and for reasons given in comments 3 and 4, a NN is better suited to this application compared to a decision tree algorithm.

Using a continuum classification could be useful but changes the filtering from the "good" or "bad" binary classification which is trying to filter out retrievals where the forward model is suboptimal to an expected precision requirement. A continuum classification would result in "good" retrievals being filtered out because they don't meet the precision requirements due to higher than expected variance for certain scenes. For example, if the user wanted to filter for retrievals with a precision of better than 1 ppm, they should expect that they would not get any retrievals over snow, which probably has a precision of less than 1 ppm, even though the retrievals that they are filtering out are good. Moving towards a continuum classification changes the problem from trying to eliminate bad retrievals to filtering for retrievals that meet precision requirements, which could be useful depending on the application.

In recognition that the discussion is lacking in an explanation of choices made with the NN, potential improvements to the algorithm, drawbacks of using a NN and use of other algorithms the following text was added on lines 318-339:

"The main downside to using a neural network to filter OCO-2 retrievals is that it doesn't readily provide information on why a retrieval was classified as "good" or "bad" which would be useful for improving the retrieval algorithm. Decision tree algorithms are binary classifiers which do provide information on the classification of data. However, a draw-back to decision trees is that they over fit the training data. In section 3, it was explained why some of the training data might be incorrectly classified leading to the NN filter determining an ambiguous classification for some retrievals. The assumption with decision trees is that all the training data is correctly classified, which is not entirely true in this case. The NN calculating values that make the classification of some retrievals ambiguous can be interpreted as a confidence in the classification of retrievals. With a decision tree, there is no metric to measure the confidence in the classification of retrievals so you could get retrievals passing the decision tree filter that are actually closer to "bad" retrievals than "good"".

There are also ways that the implementation of a NN to filter OCO-2 retrievals can be improved upon. Increasing the amount of retrievals used in the training data set would improve the performance of the NN filter. This can be done by incorporating new coincident TCCON measurements from the sites in this study, retrievals from new TCCON sites coming into operation and the potential use of other similar truth proxies such as the COllaborative Carbon Column Observing Network (COCCON) (Frey et al. 2018). To improve the training classification, other helpful truth proxies such as cloud and aerosol information can be combined with $XCO_2{}^{Diff}$ when classifying the retrievals for training. The current implementation of the NN

as a binary classifier was done to make the problem as simple as possible in order to filter out retrievals where the forward model of the retrieval algorithm is suboptimal, rather than scenes of high variance. A possible alteration to the algorithm would be to do the classification on a continuum where the output of the NN ($\hat{Y}$) would be related to the expected precision of the data. The downside to this configuration would be that the NN filter would be filtering out not only bad retrievals but also scenes with high variance. For example if you wanted a precision of better than 1 ppm chances are you would not have any retrievals over snow getting through the NN, greatly reducing throughput in the winter and shoulder season at high latitudes."

*Minor Comment 1 - L147: Was there any attempt to optimize the features being fed into the algorithm?*

Yes there was some attempt to optimize the feature list in order to save some computer time when training the NN. Initially we started with a feature list that contained all features used in the OCO-2 qc_flag filter and in the retrieval state vector. Additional features that provided information on airmass, quality of the spectral fit, and quality of the spectrum were also included. Features that seemed to provide similar information to other features were removed from the feature list. This was done by training the NN with and without the feature and assessing the importance of the feature by looking at the bias, precision, amount of outliers ($XCO_2^{Diff} > 2.5$ppm) getting through and throughput.

The following text was added as lines 148-154 to clarify this:
"An initial feature list was built by combing all features in the OCO-2 qc_flag filter (Osterman et al., 2020), with the features contained in the retrieval state vector. Features that provide information about the quality of the spectral fit, the quality of the recorded spectrum, and airmass were also included in the initial features list. To reduce the total number of features used, each feature that was thought to provide redundant information to others was removed by testing how the NN performed with and without the feature. The bias, precision, number of outliers (absolute value of $XCO_2^{Diff} > 2.5$ppm) getting through and throughput of the training data set were used as the metrics to judge the NN performance with and without the feature."

*Minor Comment 2 - L205: How is the threshold of 0.1 determined?*

The threshold of 0.1 was determined by looking at the precision, number of major outliers (absolute value of $XCO_2^{Diff} > 2.5$ppm) and throughput at different threshold values. Fig. 2 shows that as you increase the threshold you will gain some throughput but at the same time, the spread in the data increases and as one approaches Y = 0.2, you get retrievals that pass the NN filter that have a bias ($XCO_2^{Diff}$) of ~20 ppm. The 0.1 threshold was adopted because it gives a good balance between increase in throughput, improvement in precision, and limiting the major outliers getting through the NN filter. From a user point of view, depending on the application, one can always adjust this threshold value to increase the throughput while degrading precision or decrease throughput while improving precision.

The following text on lines 211-213 was added:

"This threshold of 0.1 was determined by trying to balance throughput with degradation of precision as well as limiting the amount of individual retrievals with high absolute $XCO_2^{Diff} > 2.5$ ppm passing the NN filter."

*Minor Comment 3* - L214: I do worry a bit that so many small values of $XCO_2^{Diff}$ have values of $\hat{Y} > 0.1$. It may be true that the greatest density of points with low $XCO_2^{Diff}$ exists at $\hat{Y} < 0.1$, but if you integrate along the $XCO_2^{Diff} = 0$ line from $\hat{Y} = 0.2$-$1.0$ (in Fig. 3b), you get a non-insignificant portion of the total samples that should be considered "accurate," according to the TCCON data. Could you discuss the implications? If this means that the NN filter is being overly restrictive, I just wonder if this justifies a re-framing of the problem, re:the first comment, to use a different ML algorithm or change to a non-binary determination.
Fig. 3: Should $XCO_2^{Diff}$ have units of ppm? It would be helpful if this were indicated.

Also, if 2.5 ppm is the threshold for deeming a value of $XCO_2^{Diff}$ as "good" or "bad," it would make sense to indicate those values (+/- 2.5) as horizontal lines in both panels.

Likewise, since $\hat{Y}$ values are considered good if <0.1, bad if >0.1, indication of that threshold as a vertical line would also be helpful.

The NN calculates an ambiguous classification (i.e $\hat{Y} > 0.0$ and $< 1.0$) when there is no clear distinction in the amount of "good" and "bad" examples of retrievals with similar feature values in the training data set (illustrated in comment 4). This could happen when you have a scene of high variance and no clear majority between "good" and "bad" due to lack of data. For example, let's say you had a snow scene with only 10 retrievals with similar feature values and roughly equal "good" and "bad" classifications. This is possible because the low albedo of snow (low signal intensity) causes high variance and it just so happens that 5 retrievals have absolute value of $XCO_2^{Diff} < 2.5$ ppm and the other 5 have an absolute value of $XCO_2^{Diff} > 2.5$ ppm. The NN would get 5 retrievals that need a value of 0 and the other 5 need a value of 1.0, so a value of 0.5 would decrease the cost function. In this case having roughly similar amounts of "good" and "bad" retrievals will make the classification ambiguous. The second reason is that $XCO_2^{Diff}$ is not a perfect proxy for classification (as discussed in major comment 1). In this case some portion of retrievals with similar feature values are incorrectly classified resulting in an ambiguous classification. To err on the side of caution, the retrievals with an ambiguous classification have been classified as bad because we do not have enough confidence in their classification.

The following text on line 225-233 was added:

"Clearly there are retrievals with an ambiguous classification ($\hat{Y} > 0$ and $< 1.0$) even though all retrievals in the training data set were assigned a value of 0 or 1. For the NN to achieve an ambiguous value when training there would have to be retrievals with similar feature values, with no clear majority between "good" and "bad" examples in the training data. This could happen because $XCO_2^{Diff}$ is not a perfect classification metric which could lead to some portion of retrievals being incorrectly classified. Another possibility is a lack of data combined with the real variance in the scene could result in the no clear majority case leading to an ambiguous classification. Since the NN is not showing any confidence in the classification of these

retrievals, they are manually classified as "bad" to err on the side of caution. There is also the possibility that actual bad retrievals can get through the NN filter due to insufficient training examples as well as incorrect classification of training data."

For fig. 2, we have now added units of ppm on the x-axis of both panels. On panel b, of the same figure, dashed lines were added to indicate ±2.5 ppm. On fig. 3, a dashed line was added to both panels at 0.1 to indicate the boundary between "good" and "bad", and ppm units were added to the label of the y-axis.

Minor Comment 4 - L216: The influence of the proportion of "good" vs. "bad" training data could be tested, granted there are enough data available. Subsample the good retrievals so as to still be representative of various conditions (perhaps sample across percentile bins of each input feature) and allow roughly equal numbers of good points as bad points. It would be interesting to see how the results change; this is likely not be the best way to set up the NN training but could be illustrative.

Due to the number of features and limited amount of data to train, it was not possible to acquire enough data to do what was requested by the reviewer. Instead, only four features were selected with the data filtered for a limited range of values (chosen to maximize sample size) to get retrievals with similar feature values. Instead of using $XCO_2^{Diff}$ to classify the training data, the retrievals were assigned a value of 0 ("good") or 1 ("bad") randomly but with equal amounts of good and bad data. The NN was trained with this data set (for the four features) and the final $\hat{Y}$ was clustered at ~0.5 as shown in the extra figure 1. This makes sense because the data has similar feature values but half the data was assigned 0 and the other half 1, which means that $\hat{Y} = 0.5$ will reduce the cost function. This was repeated for unequal amounts "good" and "bad" examples in the training data set and the cluster around $\hat{Y}$ shifted accordingly, which is consistent with what was described in the previous comments. Therefore, for retrievals with similar feature values, if there is not a clear majority between "good" and "bad" examples in the training data, then the classification will end up being ambiguous.

[Figure]

Extra Figure 1. The NN calculated value for a subset of retrievals with similar feature values. The NN was trained with only four features and for each retrieval a classification of "good" or "bad" was randomly assigned to have equal amounts of "good" and "bad" examples in the training data set.

*Minor Comment 5 -* L290: Could the authors discuss the considerations that go into the OCO-2 team's determination of the B10 qc_flag?

Along these lines, it seems as though, from this discussion and from Fig. 8, that the qc_flag may be too dependent on the presence of snow, when in fact there are other complicating factors, e.g., during summer, that should be more heavily weighted when checking the quality of the OCO-2 data. Meanwhile, some of the over-snow retrievals may contain better data than previously acknowledged. This could be a valuable contribution to the field if the authors agree this conclusion is supported by their analysis. If the lack of independent observations precludes confidence in this supposition, then please disregard.

As described fully in O'Dell et al. (2018) for a previous version of the data, the OCO-2 B10 qc_flag is built by identifying the parameters that most strongly correlate with bias as well as scatter in the retrieved $XCO_2$. In general, these variables typically relate strongly to surface albedo, retrieved aerosol parameters, retrieved surface pressure, retrieved $CO_2$ profile shape, signal-to-noise ratios, and fit quality (in terms of reduced $X^2$ values and relative RMS residuals). Snow and ice-covered surfaces are mostly filtered out as well, because these surfaces have extremely low values of signal in the weak and strong $CO_2$ bands. Further, these surfaces are

thought to induce significant biases in the retrieved $XCO_2$ due to the poorly modeled surface in the forward model of the retrieval algorithm.

Yes, we agree this analysis does show that the OCO-2 B10 retrieval algorithm does show potential of working over snow covered scenes. However, we think that there needs to be more work done on this topic to understand the conditions under which the retrieval works for snow. In this study there are only 3219 OCO-2 soundings coincident with TCCON data (in the validation data set) that are snow scenes as classified by the OCO-2 snow_flag. More data would be preferred to get good enough site statistics of bias and precision over snow.

The following text on line 367-377 were added:

"The NN filter passes some retrievals of soundings made over snow albeit at a lower throughput compared to non-snow scenes. This is expected because retrievals over snow are difficult due to the low albedo in the spectral regions of the $CO_2$ bands, often providing insufficient signal for a good retrieval. However, the albedo of snow is dependent on the age of the snow with fresh snow having higher albedo compared to old snow, so there is a possibility that some of the soundings recorded over snow have enough signal to produce a good retrieval. Nevertheless, these retrievals are further complicated by the fact that the spectra are usually recorded through large solar zenith angles (SZA) due to the soundings being made at either high latitudes or at times of the year where the SZA is large which is challenging for the radiative transfer model of the retrieval algorithm to deal with. The results of this study show the potential of a machine learning algorithm to tease apart these factors and recover some of the retrievals over snow, although in this study there wasn't enough coincident data over snow to get meaningful site statistics (bias and precision). A future study will investigate the potential of a machine learning algorithm to filter the retrievals over snow by folding in more training and validation data."

*Minor Comment 6* - L325: Another potential future direction I would offer is performing bias correction on the retrieved $XCO_2$. Recent modeling studies have moved in this direction, with forecasts of surface air quality being adjusted on a site-specific basis using machine learning and observations (e.g., https://acp.copernicus.org/articles/20/8063/2020/acp-20-8063-2020.html and https://acp.copernicus.org/articles/21/3555/2021/acp-21-3555-2021.html). It seems this approach may be applicable to satellite-retrieved species with available independent measurements.

The OCO-2 $XCO_2$ retrievals have a global bias correction derived from retrievals coincident with all TCCON sites as detailed in Osterman et al. (2020). It is possible to use a machine learning algorithm to develop a regional bias correction using TCCON data, however, one possible problem is that there is a limited amount of TCCON sites and so the approach won't provide any measurements for large parts of the Earth. Despite this difficulty, it could still be useful.

The following text on line 383-386 was added:

"One possible future application of the NN (or other machine learning algorithms) could be to improve the bias correction of OCO-2 retrievals. Le et al. (2020) used a convolution NN for

spatiotemporal bias correction of satellite precipitation data and air quality forecasts have moved towards bias correction using a decision tree algorithm (Ivatt and Evans, 2020)."

*Technical corrections:*

L299: Instead of "topography," perhaps "variable topography" would be more clear?

Table 1, 4th- and 3rd-to-last rows: "form" should be "from"

Fig. 3 caption: "the all three" should be "all three"

All technical corrections were fixed.

---

## Author Comment (AC2)

**Assessing the Feasibility of Using a Neural Network to Filter OCO-2 Retrievals at Northern High Latitudes**

Referee comments are given in Blue.

Responses to comments are given in Black.

In the manuscript highlighted text is the added text and  text is deleted text.

**Response to Referee 2**

We thank the reviewer for the comments on our manuscript. Please see below for our responses.

Major Comment 1 - My only significant criticism on the method is the use of 0/1 quality parameters. Indeed (line 173) the target for the neural network is 0 when the XCO2 error is less than 2.5 ppm and 1 when it is larger than that. This means that a sounding with an error of 2.45 is considered as good as a sounding with an error of 0, while a sounding with 2.55 is as bad as that with an error of 7 ppm. I would have suggested to rather train the NN with a target that is a continuous function of the absolute error |TCCON-OCO2].

Similarly, I am surprise by the choice of the threshold at 01 (line 205) that is not justified. It would have been interesting to show the the standard deviation of the error as a function of the NN output (before the 0/1) classification. This would have provided arguments for the selection of the selection threshold (currently set at 0.1)

Training the NN with a continuous function of the absolute error between TCCON and OCO-2 would change the question from is the retrieval "good" or "bad" (in other words when does the retrieval algorithm work well vs when is it lacking) to a precision requirement. If the target is a precision requirement, the NN will filter out both "bad" retrievals as well as "good" retrievals where the signal is not good enough to achieve the required precision. This will lead to specific scene types (i.e snow) being filtered out which would lead to less soundings in the shoulder seasons. As long as this is understood this could be a viable path forward to augment the algorithm. The following was added on lines 333 - 339 to discuss this:

"The current implementation of the NN as a binary classifier was done to make the problem as simple as possible in order to filter out retrievals where the forward model of the retrieval algorithm is suboptimal, rather than scenes of high variance. A possible alteration to the algorithm would be to do the classification on a continuum where the output of the NN ($\hat{Y}$) would be related to the expected precision of the data. The downside to this configuration would be that the NN filter would be filtering out not only bad retrievals but also scenes with high variance. For example if you wanted a precision of better than 1 ppm, it is likely that you would not have any retrievals over snow getting through the NN, greatly reducing throughput in the winter and shoulder season at high latitudes."

The boundary set for the classification of the training data (2.5 ppm) does not translate to a hard boundary. As per the discussion from the comments (major comment 1 as well as minor comments 3 and 4) made by Referee 1, the way the NN works is by a majority determination from the training data. This means that after training the NN there could be retrievals that pass

the NN which have an absolute $XCO_2^{Diff} > 2.5$ ppm as long as the majority of retrievals with similar feature values have an absolute $XCO_2^{Diff} < 2.5$ ppm. Fig 3a shows that there are retrievals with absolute $XCO_2^{Diff} \sim 0$ ppm that don't pass the NN filter, which is attributed to no clear majority of retrievals with similar feature values that were classified as "good" in the training dataset.

The threshold of 0.1 was determined by not only looking at the standard deviation of the error as a function of the NN output (before manually setting it to 0 or 1), but also the number of major outliers and throughput at different threshold values (i.e. $\hat{Y}$). Fig 3a gives an indication of the standard deviation as a function of $\hat{Y}$ and also shows that as $\hat{Y}$ approaches 0.2 there are outliers as high as 20 ppm. Fig 3b also gives an indication of the standard deviation as a function of $\hat{Y}$ as well as the throughput for a given value of $\hat{Y}$. Setting a threshold value of 0.1 was found to give a good balance between increases in throughput, improvement in precision, and limiting the amount of major outliers getting through.

The following text on lines 211-213 was added:
"This threshold of 0.1 was determined by trying to balance throughput with degradation of precision as well as limiting the amount of individual retrievals with high absolute $XCO_2^{Diff} > 2.5$ ppm passing the NN filter."

Minor Comment 1 – The abstract could mention the NN input features that seem to have the highest influence on the results

Extracting which features have the biggest influence on the entire coincident data set is not possible in the way that was done to show why the NN filtered out most of the data at Eureka. This is because when looking at the entire coincident data set there are a lot more features that become important because of different geophysical scenes provided at the different sites.

Minor Comment 2 – Some of the technical description of the NN approach (lines 150-160, lines 186-196) may not be needed in the paper as they are described in other documents

The authors believe that all essential details used to derive the results should be included in the manuscript so that anyone can reproduce the results given just the equations in the manuscript and supplementary material in combination with the OCO-2 and TCCON data sets. Also, this manuscript will serve as the main reference for future manuscripts that will deal with changes to the NN algorithm as well as using other training datasets.

Minor Comment 3 – Line 219: I do not think that "separated into two datasets" is appropriate as some elements of the original dataset are in none of the two while some others are in both

We see how this line can be misinterpreted. The following line "To validate the NN filtering, the validation data set was separated into two data sets. One data set was the OCO-2 bias-corrected XCO₂ values filtered using the NN filter and the other was filtered using the B10 qc_flag=0." was taken out and replaced with "To validate the NN filtering, the NN filter was applied to the validation data set and compared to the same validation data set but with the B10 qc_flag=0 applied to the soundings." on lines 236-238.

Minor Comment 4 – The paper shows the pass fraction as a function of 3 of the NN input parameters (features) (Figure 4). I assume the same has been done for the others. If not mentioned, I assume it means there is so significant variation.  Please confirm

Yes, the same analysis and plots were done for all features. It was found that there was no significant variation for the rest of the features.

Minor Comment 5 – Figure 3a is impossible to read as they are two many datapoints. I strongy suggest to change the figure style, or to make a random sample of the datapoints before ploting

For fig 3, a red dashed line was added at $\hat{Y}$ =0.1 so that the reader can better compare the data that is considered "good" vs "bad". Other than that the figure was kept the same because the reader needs to be able to see how many major outlies (as well as their magnitude) gets through the NN filter as $\hat{Y}$ increases. This is important for the reader to see as this helps to illustrate the reason why the 0.1 threshold value was selected.

Minor Comment 6 – Figure 5-7 are difficult to read. I believe it would be better by showing two bars side my side rather than the plain/dash drawing

Figures 5-7 were plotted with the plain/dash bars side by side rather than over lapping (see below). The authors agreed that this way of plotting the figures did not enhance the clarity of the figures and find the overlapping bars easier to interpret. So figures 5-7 in the manuscript are unchanged.

[Figure]

Figure 5 The bias at each site for the different seasons when the NN filter (bars with solid lines) and qc_flag (bars with dashed lines) is used to filter the OCO-2 retrievals in the validation data set. a) Spring (March, April, May). b) Summer (June, July, August). c) Fall (September, October, November). d) Winter (December, January, February). Note the different y-axis ranges for each plot. Note that bars that show a bias of zero are due to no data rather than a bias of zero.

[Figure]

Figure 6 Same as Figure 5 but shows the precision at each site for each season. a) Spring (March, April, May). b) Summer (June, July, August). c) Fall (September, October, November). d) Winter (December, January, February). Solid bars indicate the NN filter, and dashed bars indicate the original B10 qc_filter.

[Figure]

Figure 7 Same as Figure 5 but shows the number of soundings that pass each filter at each site for the different seasons. a) Spring (March, April, May). b) Summer (June, July, August). c) Fall (September, October, and November). d) Winter (December, January, February). Solid bars indicate the NN filter, and dashed bars indicate the original B10 qc_filter.